# First Report of a Psyllid Vector of ‘*Candidatus* Phytoplasma pruni’ (Strain 16SrIII-J)

**DOI:** 10.3390/plants14091279

**Published:** 2025-04-23

**Authors:** Tomás Llantén, Sebastián Cabrera, Javiera Fuentes, Camila Gamboa, Constanza González, Alan Zamorano, Tomislav Curkovic, Daniel Burckhardt, Nicola Fiore

**Affiliations:** 1Departamento de Sanidad Vegetal, Facultad de Ciencias Agronómicas, Universidad de Chile, La Pintana 8820808, Chile; tomas.llanten@ug.uchile.cl (T.L.); sebacabrerahijo@gmail.com (S.C.); javiera.fuentes.1@ug.uchile.cl (J.F.); camila.gamboa@uchile.cl (C.G.); c.gonzalezlevinir@gmail.com (C.G.); agezac@uchile.cl (A.Z.); tcurkovi@uchile.cl (T.C.); 2Programa de Magíster en Ciencias Agropecuaria, Facultad de Ciencias Agronómicas, Universidad de Chile, La Pintana 8820808, Chile; 3Naturhistorisches Museum, 4001 Basel, Switzerland; daniel.burckhardt@bs.ch

**Keywords:** *Russelliana solanicola*, *COI* gene, Chile, mallow psyllid

## Abstract

In Graneros, O’Higgins Region, Chile, the mallow psyllid (*Russelliana solanicola* Tuthill, 1959) from *Malva nicaeensis* L. was identified as a potential vector of ‘*Candidatus* Phytoplasma pruni’. Over an 8-month period, 2089 specimens of a species of Psylloidea, including immatures and adults, were captured. We only selected the adults used for transmission trials in *Catharanthus roseus* (L.) G. Don (periwinkle) plants. By nested PCR, using primer pairs for phytoplasma detection in *16S rRNA* and *IdpA* genes, 7 out of 113 (6.2%) periwinkle plants used in transmission trials were found to be infected by phytoplasmas. Insects that fed on these plants also tested positive for the same phytoplasmas. Periwinkle plants never showed virescence and phyllody, as commonly occurs with phytoplasma 16SrIII-J infection due to the effector SAP54. In this case, using primer pairs for the *SAP54* gene, an amplification product was never obtained. Virtual restriction fragment length polymorphism (RFLP) analysis of F2nR2 fragments indicated that the phytoplasma, found in both periwinkle plants and insects used in transmission trials, belongs to the 16SrIII-J ribosomal subgroup. The *COI* gene of the psyllids samples was amplified and sequenced, showing a similarity ranging from 84.84% to 85.02% with *R. solanicola* from *Solanum tuberosum* L. The mitochondrial genome of the psyllid was also sequenced, revealing a 14,835 bp circular DNA molecule with 37 genes. The mallow psyllid transmitted the phytoplasma 16SrIII-J to periwinkle plants. The molecular identification of the insect does not match the morphological one, indicating that the mallow psyllid may constitute a cryptic species within the polyphagous *R. solanicola* species. This is the first report of a psyllid as a vector of the phytoplasma 16SrIII-J.

## 1. Introduction

Phytoplasmas are parasitic microorganisms belonging to the domain bacteria, class Mollicutes, and are assigned to the provisional genus ‘*Candidatus* Phytoplasma’ (‘*Ca*. P.’) due to the impossibility of confirming Koch’s postulates [1].

The 16S rRNA gene supports the monophyly of phytoplasmas within the Mollicutes. This gene, proposed as a universal marker by Pace et al. [2], is found in all organisms and is essential for constructing phylogenetic relationships between bacteria [1,3].

Phytoplasmas are wall-less bacteria that live in the phloem of plants and in the tissues of insects. They infect over a thousand plant species and are associated with numerous diseases, many of which cause significant damage, especially in woody plants [1,4]. They spread through infected plant material, seeds, and animal vectors [5]. In insect vectors, phytoplasmas concentrate in the salivary glands, multiply, and persist until the insect dies. Most phytoplasma vector species belong to the Hemiptera, in particular suborders Auchenorrhyncha (Cicadellidae, Cixiidae) and Sternorrhyncha (Psylloidea).

Psyllids (Hemiptera: Psylloidea) with a body length between 1 and 10 mm comprise slightly over 4000 described species worldwide, which are assigned to seven families. They differ from other Sternorrhyncha in adults, which are always winged in both sexes and which bear hind legs modified for jumping, and also have generally very narrow host ranges [6]. Host plants belong mostly to eudicots and Magnoliales as well as, to a much lesser extent, monocots and conifers [6,7,8]. Psyllids usually feed on phloem. Within the Aphalaroidinae (Psyllidae), *Russelliana* Tuthill, 1959, is, with 43 described species, the largest genus [9,10]. Species of the genus *Russelliana* are monophagous or oligophagous, apart from one, *Russelliana solanicola* Tuthill, 1959, which is considered polyphagous. Serbina et al. [11] reported plants of at least ten different families, four of which have been confirmed as hosts by the presence of immatures.

Vectors acquire phytoplasmas by feeding on the phloem of infected plants and subsequently transmit them. The phytoplasma–insect vector relationship is divided into three phases: acquisition, latency, and transmission. In the acquisition period, phytoplasmas enter the insect through the stylets while socking phloem sap of infected plants. During the latency period, the bacteria move through the digestive tract and settle in the salivary glands. Only from this point can the insect transmit phytoplasmas to healthy plants, inserting saliva into them as the first step in feeding. The vector maintains infectivity throughout its life as the phytoplasma multiplies inside it [12,13].

Many non-cultivated plant species (mainly weeds) serve as reservoirs for phytoplasmas, including *Amaranthus retroflexus* L., *Calystegia sepium* L., *Cirsium arvense* L., *Convolvulus arvensis* L., *Datura stramonium* L., and *Urtica dioica* L. [14]. In Chile, phytoplasmas have been identified in the aforementioned species as well as in *Polygonum aviculare* L., *Galega officinalis* L., *Brassica rapa* L., *Rubus ulmifolius* L., *Rosa* spp., *Malva* spp., and *Erodium* spp. [15,16].

The presence of phytoplasmas in plants causes hormonal imbalance and alterations in photosynthesis and reserve accumulation. These effects result in symptoms such as yellowing, the reddening of leaves and veins, floral sterility, virescence, stunting, vegetative disorders, leaf curling, decline, phyllody, and witches’ broom [17]. Although symptoms may vary between plant species, different phytoplasmas can cause similar symptoms in different plant species [18].

In Chile, several phytoplasmas have been detected in different plant species of agricultural interest. In the Biobío region, ‘*Ca.* P. pyri’ was found in pear plants (*Pyrus communis* L.), with Pear Decline symptoms [19]. In vineyards with Grapevine Yellows, phytoplasmas 16SrI-B, 16SrI-C (‘*Ca.* P. asteris’), 16SrVII-A (‘*Ca.* P. fraxini’), 16SrXII-A (‘*Ca.* P. solani’), 16SrIII-J (‘*Ca.* P. pruni’), and 16SrV-A (‘*Ca.* P. ulmi’) were found [20,21]. ‘*Ca.* P. pruni’ was detected in lettuce and chard crops [22]. Additionally, the presence of phytoplasmas from ribosomal subgroups 16SrXIII-F (‘*Ca.* P. hispanicum’) and 16SrV-A was confirmed in citrus plants in the Metropolitan and Libertador Bernardo O’Higgins regions, respectively [23]. ‘*Ca.* P. hispanicum’ was also detected in strawberry plants [24]. Potential ‘*Ca.* P. pruni’ (strain 16SrIII-J) vectors have been reported, such as *Amplicephalus curtulus* Linnavuori & DeLong*, Amplicephalus ornatus* Linnavuori, *A. pallidus* Linnavuori, and *Exitianus obscurinervis* Stål (Cicadellidae). These insects, captured in the Valparaíso, Libertador Bernardo O’Higgins, and Maule regions, mainly pass their biological cycle on weeds (that behave as phytoplasma reservoirs) and feed on grapevines (or several other crops) only occasionally, transmitting the phytoplasmas present in weeds [16,25]. To confirm if an insect species is a vector of phytoplasmas, transmission trials are conducted. In Chile, it has been demonstrated that the leafhoppers *Paratanus exitiousus* (Beamer) and *Bergallia valdiviana* Berg are vectors of the phytoplasma 16SrIII-J. These assays were performed on the indicator plant *Catharanthus roseus* (L.) G. Don (periwinkle) and *Vitis vinifera* L. cv. Cabernet Sauvignon [26]. Furthermore, the phytoplasma 16SrIII-J, which causes symptoms such as leaf curling, witches’ broom, and yellowing, has been detected in potatoes and other economically important plant species [27].

Fuentes [25] reported for the first time that the mallow psyllid is a potential vector of the phytoplasma 16SrIII-J. Considering that this phytoplasma is commonly associated with leafhopper, its presence in psyllids is extremely interesting.

Based on this information and considering that the spread of a phytoplasma and the damage it causes are closely related to the transmission efficiency of its insect vectors, it is important to clarify whether the mallow psyllid is a vector of phytoplasma 16SrIII-J.

## 2. Results

### 2.1. Psyllid Collection and Morphological Identification

From mallow (*Malva nicaeensis* L.) plants, were captured 2089 specimens of mallow psyllid (Figure 1), including immatures and male and female adults, and they were morphologically identified as *R. solanicola*.

### 2.2. Transmission Trials

A total of 113 transmission trials were conducted, involving the use of periwinkle plants and adult individuals of the mallow psyllid (Figure 2). Table 1 summarizes relevant information regarding the captured individuals, the number of captures by date, and the number of transmission trials initiated on each date.

### 2.3. Phytoplasma Detection

The 113 periwinkle plants used in the transmission trials were analyzed monthly for a year, and we found 7 samples (6.2%) that successfully amplified the *16S rRNA* gene. Seven months after the start of the transmission trials, no more phytoplasma-positive plants were obtained (Table 2). All the sequences obtained with nested PCR in *16S rRNA* gene (R16F2n/R2; 1200 bp) matched, in BLASTn, with *sugar beet yellow wilt disease phytoplasma* (GenBank accession number KM658257.1), a phytoplasma of the 16SrIII-J ribosomal subgroup [21], showing a nucleotide sequence identity range of 99.82% to 100% (Table 2).

Using phylogenetic analysis, all samples were grouped within the same clade, which included the reference sequence of the phytoplasma belonging to the 16SrIII ribosomal group (Figure 3). The information displayed in the phylogenetic tree constructed using both the maximum likelihood and neighbor-joining methods is consistent, with both approaches showing the same clades and associations between species sequences and reference sequences.

Nested PCR amplification of phytoplasma 16S rDNA was performed on the insects that fed on the 7 phytoplasma-positive periwinkle plants and on 13 other psyllids samples used in transmission trials. Nucleotide sequences of all the PCR fragments were the same (Genbank accession number PV053119) and shared 99.82% identity with the sequence of the *sugar beet yellow wilt disease phytoplasma* strain.

The virtual RFLP pattern derived from the F2nR2 fragment of the 16S ribosomal DNA from phytoplasma-infected periwinkle plants (GenBank accession numbers: PV053118, PV053117, PV053116, PV053115, PV053114, PV053113, PV053112) and psyllids (GenBank accession number: PV053119) turned out to be identical (similarity coefficient 1.00) to the reference pattern of group 16SrIII, subgroup J (GenBank accession number: AF147706). The phytoplasma under study is a member of 16SrIII-J. When all available restriction enzymes were applied, we obtained the same digestion pattern as was derived from RFLP analysis of the reference sequence *Chayote witches’ broom phytoplasma* ChWBIII (Ch10) [28]. The fragments subjected to in silico RFLP analysis (figure not shown) exhibited profiles identical to those available for phytoplasmas enclosed in ribosomal subgroup 16SrIII-J, using *sugar beet yellow wilt disease phytoplasma* for reference [21].

Amplification products were obtained from all phytoplasma-positive periwinkle plants and the corresponding psyllids used in transmission trials through the use of specific primers for the detection of the immunodominant membrane protein (*IdpA*). The resulting sequences were identical for each sample and BLASTn analysis indicated that the closest phytoplasma was *Bellis virescence* (BellVir—GenBank accession number MG435349.1), with 96.86% identity. BelVir belongs to the 16SrIII-J ribosomal subgroup [29], a result that matches the findings from the analysis of the *16S rRNA* gene sequences.

### 2.4. Detection of the SAP54 Effector

Given the presence of effectors such as the SAP54 homolog (AYWB_224) in the Chilean strain of the 16SrIII-J ribosomal subgroup phytoplasma [30], it was expected that phyllody would develop in infected plants over time. However, the plants did not show phyllody (Figure 4). PCR was performed using specific primers to amplify the gene encoding of the SAP54 effector to determine whether this effector was present in the plants, as well as in the psyllids. Negative results indicated that the effector SAP54, which is an important component of the pathogenicity mechanism of the 16SrIII-J phytoplasma [30,31], was not present in any of the phytoplasmas detected in psyllids and in the infected periwinkle plants.

### 2.5. Molecular Identification of Psyllids

All the insects used in transmission trials successfully amplified the mitochondrial *COI* gene region and showed nucleotide sequence identities ranging from 99.8% to 100% among them. According to Zhang and Bu [32], in the order *Insecta*, an individual is considered to belong to the same species as the reference when the *COI* gene nucleotide sequence identity is ≥97%. Serbina et al. [33] suggested that, in some species of the psyllid genus *Melanastera*, species limits may be lower. Therefore, it is possible that the specimens collected and analyzed in this study belong to an insect species in which the *COI* gene is not present in the genomic databases. The closest mallow psyllid species is *R. solanicola* with nucleotide similarity ranging from 84.84% to 85.02% (Table 3). This means that the mallow psyllid is genetically distant from the genus *Russelliana*, whose species form a separate clade containing the *COI* gene sequence of two individuals, *Russelliana* sp. (GenBank accession number MG988824) and *R. solanicola* (GenBank accession number NC_038140.1) (Figure 5).

For the insect mitochondrial genome, total DNA sequencing yielded a total of 17,092,516 reads, which were de novo assembled, resulting in a total of 230,752 contigs or consensus sequences. The selection of the contig containing the mitochondrial genome was performed using a BLAST-based search tool incorporated into the CLC Genomics Workbench software, considering an approximate size of 16,000 bp. A single contig with a consensus size of 14,835 bp was identified, composed by 78,806 reads, reaching a coverage of 795.11×. The expected size of approximately 16,000 bp was established based on the mitochondrial genome of *R. solanicola* available in GenBank.

Mitogenome annotation retrieved 13 genes (*NAD2*, *COX1*, *COX2*, *ATP8*, *ATP6*, *COX3*, *NAD3*, *NAD5*, *NAD4*, *NAD4L*, *NAD6*, *CYTB*, and *NAD1*), 22 transfer RNAs (*tRNAI*, *tRNAQ*, *tRNAM*, *tRNAW*, *tRNAC*, *tRNAY*, *tRNAL2*, *tRNAK*, *tRNAD*, *tRNAG*, *tRNAA*, *tRNAR*, *tRNAN*, *tRNAS1*, *tRNAE*, *tRNAF*, *tRNAH*, *tRNAT*, *tRNAP*, *tRNAS2*, *tRNAL1*, and *tRNAV*), the ribosomal RNA operons (*rRNAL* and *rRNAS*), and a control region with a high A-T content (78.91%).

The complete mitochondrial genome of the mallow psyllid corresponds to a circular double-stranded molecule, 14,835 bp in length (Figure 6). It is composed of 37 genes, including 13 protein-coding genes (PCGs), 22 tRNA genes, 2 rRNA genes, and one control region. Among the 37 genes, the J strand encoded 9 protein genes (*NAD2*, *COX1*, *COX2*, *ATP8*, *ATP6*, *COX3*, *NAD3*, *NAD6*, and *CYTB*) and 14 tRNA genes (*tRNA-Ile*, *tRNA-Met*, *tRNA-Trp*, *tRNA-Leu2*, *tRNA-Lys*, *tRNA-Asp*, *tRNA-Gly*, *tRNA-Ala*, *tRNA-Arg*, *tRNA-Asn*, *tRNA-Ser1*, *tRNA-Glu*, *tRNA-Thr*, and *tRNA-Ser2*), while the N strand encoded 4 other protein genes, 8 tRNA genes, and 2 rRNA genes (*16S rRNA* and *12S rRNA*). A total of 11 gene junctions in the genome exhibited overlapping regions, totaling 42 base pairs, which might be associated with the compaction of the mallow psyllid’s mitochondrial genome [34].

After comparing the sequences of the identified genes with the closest reference species in the GenBank database using BLASTn, 13 common protein-coding genes were identified between the two psyllid species. A notable characteristic is that all these genes share nucleotide identity similarities below 84% (Table 4).

Table 5 presents the annotations for the mitochondrial genome of the mallow psyllid, detailing the gene name, its position within the mitochondrial genome, its size (bp), its start and stop codons, and its number of intergenic nucleotides.

The proposed mitochondrial genome of the insect vector (GenBank accession number PV083681) was compared using BLASTn, obtaining a high coverage percentage (98%) and low nucleotide sequence (80.6%) identity with an individual of *R. solanicola* (Genbank accession number NC_038140), confirming what was observed with the *COI* gene.

The obtained sequence of the mitochondrial genome of the insect vector, provisionally named mallow psyllid, was grouped in a clade close to the mitochondrial genome of *R. solanicola*. However, phylogenetic analyses revealed a significant genetic distance between the two sequences, suggesting that they belong to different species. The phylogenetic tree presented in Figure 7, constructed using the maximum likelihood method, supports this analysis, showing that although both species share a common ancestor, the genetic divergence between them is significant enough to propose that they are not the same species.

## 3. Discussion

Morphological identification suggests that the individuals of the mallow psyllid correspond to *Russelliana solanicola*. However, analyses of the *COI* gene and mitochondrial genome do not correspond to the morphological identification. This discrepancy suggests that it is highly likely these individuals may belong to a cryptic species complex within the morphologically defined *R. solanicola*. Cryptic species are defined as two or more molecularly distinct insect species that are morphologically indistinguishable [35].

According to Serbina et al. [11], *R. solanicola* is either a single or polyphagous species or, alternatively, a complex of closely related monophagous species. Under the former concept, *R. solanicola* is the only polyphagous species of Psyllidae. In Psylloidea, polyphagy is otherwise found only in a few species of *Bactericera* (Triozidae) [11]. As polyphagy is unusual in psyllids, Serbina et al. [11] performed morphometric analysis of the various host races and found that there are no clear-cut morphological differences between the host races. Therefore, it is possible to deduce that *R. solanicola* is a single polyphagous species.

In the present study, immature and adult psyllids were found on *M. nicaeensis*, making this plant a host according to the criteria established by Burckhardt et al. [7]. This represents the first report of Malvaceae as hosts of *R. solanicola sensu lato* (*s.l.*). Molecular analysis revealed that the psyllid from mallow plants is genetically distinct from *R. solanicola* collected from *Solanum tuberosum* L. This insect species was originally described by Tuthill, 1959, from specimens collected on *Solanum tuberosum* in Peru. The molecular evidence points to the hypothesis that *R. solanicola* is a species complex that includes the mallow psyllid and a series of other species defined by their host plants.

Cryptic species complexes do exist within Psylloidea. One example is *Cacopsylla pruni*, composed of two cryptic species (A and B) [36,37,38]. On the other hand, there is also contradictive evidence between molecular and morphological data, as in *Camarotiscena* where morphologically distinct species do not differ at the molecular level (D. Burckhardt, unpublished data).

It is important to emphasize that the mitochondrial genomes of closely related species in other phyla, which comprise most of the animal diversity, regularly show enough sequence diversity to allow for their discrimination [39]. Based on the results from phylogenetic analysis and mitochondrial genome characterization, it is proposed that the mallow psyllid belongs to a complex of cryptic species within the *R. solanicola* species, linked to the transmission of phytoplasmas, specifically 16SrIII-J.

The transmission efficiency in periwinkle plants of phytoplasma 16SrIII-J by the mallow psyllid is 6.2%, which is very similar to the level observed with the leafhoppers *Paratanus exitiosus* and *Bergallia valdiviana*, with rates of 3.7% and 7.4%, respectively [26]. The other 13 psyllids batches used in transmission trials, which tested positive for the phytoplasma 16SrIII-J, did not transmit the pathogen to the periwinkle plants. The phytoplasma load in the infected insects might have been too low to ensure successful transmission. Otherwise, a hypersensitivity reaction (HR) could have been triggered in the infected plants at the point of inoculation. HR is a rapid, localized defense response in plants, activated to counteract the infectious process of pathogens or damage caused by herbivores [40]. HR could interfere with the phytoplasma’s ability to establish and multiply within the host, thereby reducing transmission efficiency, possibly because the psyllid’s saliva contains effectors recognized by the plant, triggering a pattern recognition-mediated immunity (PTI) response that limits infection [41]. Finally, the lack of transmission of the phytoplasma 16SrIII-J by the 13 psyllid batches could also be influenced by the genetic variability of the insect vector or by environmental conditions affecting the insect–pathogen relationship. A vector’s ability to transmit a phytoplasma depends on factors such as genetic compatibility between the vector and the pathogen [42]. However, further studies are needed to unravel the mechanisms regulating the insect vector–phytoplasma interaction. An additional factor that may be contributing to the lack of transmission of the phytoplasma 16SrIII-J could be the length of the latency period (LP), which is the time between phytoplasma acquisition and transmission [13]. A prolonged LP could limit the time in which the phytoplasma establishes and multiplies within the vector, resulting in a lower concentration of the pathogen available for transmission. However, previous evidence has indicated that adults may experience a shorter LP, although little is known about the temporal dynamics of phytoplasmas in this stage [43]. This low pathogen concentration could cause a lack of transmission. It is likely that recently captured adult insects introduced the phytoplasma and that, during the transmission trials, the phytoplasma was still in its LP and had not reached the salivary glands. Consequently, the phytoplasma would not have been able to transmit or be detected in the analyses performed.

The association of phytoplasma 16SrIII-J with psyllids is novel and has never been reported before, as transmission by these insects has always been linked with phytoplasmas from the 16SrX ribosomal group [44]. However, Fuentes [25] first reported that the mallow psyllid was a carrier of phytoplasma 16SrIII-J. This study reports for the first time that psyllids can also act as vectors for phytoplasma 16SrIII-J, commonly transmitted by leafhoppers, expanding the range of known insect vectors and raising new questions about the epidemiology of this phytoplasma. Molecular divergence time analysis strongly suggests that phytoplasmas, their hemipteran vectors, and host plants have been evolving together for hundreds of millions of years [45]. Phytoplasma 16SrIII-J may have developed strategies to be transmitted by psyllids, adapting to the specific characteristics of these insects. On the other hand, this psyllid may have developed mechanisms to acquire and transmit this phytoplasma, expanding its role in pathogen dissemination. The ability of phytoplasma 16SrIII-J to be transmitted by psyllids broadens the range of known vectors. This may reflect a diversification in phytoplasma transmission mechanisms, allowing these pathogens to be transmitted by a broader range of insect vectors than previously observed. The inclusion of psyllids as vectors for phytoplasma 16SrIII-J introduces new questions about the epidemiology of diseases associated with this pathogen. This discovery could influence how phytoplasma-related diseases are managed and controlled, as the role of psyllids in phytoplasma transmission and dissemination must be considered. The literature shows that leafhoppers have developed mechanisms that favor more effective transmission of this phytoplasma, suggesting that host specificity plays a crucial role in the epidemiology of phytoplasma 16SrIII-J. This phenomenon can be explained by a universal rule stating that homogeneous taxonomic groups of insects tend to transmit homogeneous taxonomic groups of pathogens, although exceptions are recognized. Thus, the nature of the vector, along with its relationship to the phytoplasma, may be crucial in the spread of the disease, inviting further investigation into the specific interactions between psyllid vectors and phytoplasmas.

The SAP54 effector is known to induce phyllody in infected plants and plays an important role in the pathogenicity of phytoplasma 16SrIII-J [31]. Its absence in both the phytoplasma strains found in periwinkle plants and psyllids could explain the lack of phyllody symptoms, suggesting significant differences between the 16SrIII-J strains transmitted by leafhoppers and psyllids. The SAP54 effector absence can be explained because of the existence of potential mobile units (PMUs). PMUs in phytoplasmas are genetic elements considered transposable or mobile. Their movement within the genome may contribute to genetic variability [46,47]. PMUs are crucial in phytoplasma genetics because they may influence the adaptation and evolution of the pathogen. Their presence and activity can affect the function of adjacent genes, potentially impacting the biology and pathogenicity of the phytoplasmas.

This finding suggests that the presence of phytoplasmas does not always correlate with the appearance of symptoms, which could complicate the detection and management of disease in crops of commercial interest. Furthermore, PMUs could be present in different configurations, which could influence the virulence of some strains. The absence of effector SAP54 also suggests that local strains of 16SrIII-J may be losing this effector or may have a non-functional variant, which could be related to the absence of symptoms and the low transmission efficiency observed. The hypothesis regarding this effector in particular gains more strength regarding PMUs when genetic composition analyses of these effectors are performed among phytoplasmas from different groups. Fernandez et al. [48] found sequences that share identities of up to 96.58% between strains of phytoplasmas belonging to subgroups 16SrIII-J and 16SrXIII-F, so this effector may be within a PMU and transferred horizontally or be absent in a phytoplasma strain.

Although periwinkle plants do not show virescence and phyllody, and presumably the SAP54 effector is absent in the phytoplasma, symptoms are observed in the plant with code TT 34, including chlorotic, yellowing leaves with necrosis (Figure 4). Necrosis could be related to HR [40]. Upon touch, the yellowing leaves exhibit a corky texture, attributed to the increased differentiation of proplastids into amyloplasts, rather than chloroplasts, due to the accumulation of reserve compounds resulting from phloem blockage caused by the phytoplasmas or as a response by the plant to them [49]. Phloem blockage makes nutrient transport difficult, leading to symptoms such as yellowing, wilting, and reduced growth [50]. This blockage could be the result of phytoplasma multiplication or the plant’s defensive response, which produces substances like callose. The blockage also leads to the overaccumulation of starch in the leaves, altering their chemical composition and texture, which makes them stiffer, rougher, and corky [49]. Additionally, phytoplasma effectors can directly attack the chloroplasts, causing their destruction and symptoms associated with the loss of these organelles in the leaves [51]. However, these mechanisms have not yet been fully studied and must be clarified to better understand the role of phytoplasmas and the symptoms they generate.

The identification of *R. solanicola* as a vector of the 16SrIII-J phytoplasma represents a significant finding in the field of phytoplasma epidemiology. Although its role in transmitting the pathogen to agricultural crops has not been confirmed, its presence in host plants suggests that it could act as a reservoir for the phytoplasma, allowing its persistence in the ecosystem. This could facilitate the subsequent acquisition and spread of the phytoplasma by other insect vectors, such as polyphagous leafhoppers, increasing the risk of infection in economically important crops. In this context, the early identification of potential reservoirs and vectors is essential for designing integrated management strategies, including preventive measures to reduce the incidence of phytoplasma-associated diseases in agricultural systems.

## 4. Materials and Methods

### 4.1. Psyllid Collection

Insects were collected in a pear orchard in the O’Higgins Region (Graneros), where mallow plants that are positive for phytoplasmas have been detected. Field visits were made approximately every 15 days over 8 months (Table 1). Insects were captured using a sweeping net. This involved quickly dragging the net over *M. nicaeensis* plants. The sampled individuals were then placed in labeled trial tubes, transported to the laboratory, and photographed.

### 4.2. Psyllid Morphological Identification

Within Psylloidea, *Russelliana* can be recognized by the combination of the following characteristics: Vertex anteriorly produced into a transverse tubercle or lobe on either side. Genal processes developed. Antenna 10-segmented; 0.8–2.1 times as long as head width; segment 3 longer than either of segments 7 or 8; with a subapical rhinarium on each of segments 4, 6, 8 and 9. Clypeus heart-shaped, weakly protruding in lateral view. Propleurites longer than their width; episternum and epimeron subequal. Metatibia 0.4–0.9 times as long as head width, without basal spine, with a crown of 4–9 ungrouped or indistinctly grouped apical spurs. Basal metatarsal segment without black spurs. Forewing widest in the middle or in apical third; 2.0–2.9 times as long as wide; pterostigma developed; vein R+M+Cu splitting into the two veins R and M+Cu; anal break close to apex of vein Cu_1b_. Male proctiger one-segmented. Identification keys to genera are available from Burckhardt (1987, 1994 and 2008) [52,53,54]. The genus *Russelliana* was revised by Serbina and Burckhardt [9], who also provided a key to species. *Russelliana solanicola* differs from all other congeners in the densely spaced surface spinules of the forewings. Morphometric analysis of the different host populations of *R. solanicola* by Serbina et al. [11] concluded that these populations are conspecific.

### 4.3. Transmission Trials

The captured individuals were observed under a LEICA S6D binocular magnifying glass. Those exhibiting the typical morphology of the mallow psyllid were used for transmission trials (Table 1). The transmission trials were conducted on phytoplasma-free periwinkle plants obtained through in vitro micropropagation and maintained in 300 mL glass jars containing 60 mL of MS medium [55]. A layer of sterile filter paper was placed on the surface of the medium to prevent the insects from feeding on the medium and ensure they only fed on the periwinkle plants. A maximum of 20 individuals were added to each jar. Each jar was sealed with gauze and cotton to prevent water condensation and insect escape. During the transmission assays, the jars were kept inside a growth chamber at 25 °C, with a 16/8 h photoperiod (light/dark). The plants in the transmission trials were observed daily to remove dead insects, which were stored in 1.5 mL Eppendorf tubes in 70% alcohol at −20 °C. The feeding period of the psyllids ended after seven days. Some 14 days after the start of the transmission trial, the plants were acclimatized for the subsequent observation of symptom development.

For the acclimatization process, the plants were transferred from the MS medium to a substrate of peat and perlite in a 2:1 ratio in 500 mL transparent plastic cups. When removing the plant from the jar, the stem area in contact with the culture medium was washed with water, and the entire plant was immersed in a liquid fungicide for 15 s (Captan and Tebuconazole). Then, the basal region of the main stem was dipped in IBA for 15 s to stimulate rooting. Once placed in the substrate, 1.5 mL of IBA at a concentration of 2000 ppm was added to the plant. Finally, the plant was covered with a 500 mL plastic cup and sealed with Parafilm. The cups were kept in growth chambers under the previously mentioned temperature and photoperiod conditions. Once the roots appeared, the upper cups were gradually removed to slowly reduce relative humidity. The estimated time for symptom development is approximately three months. Starting in the third month, from the beginning of the transmission trial, monthly analyses were conducted to detect phytoplasmas.

### 4.4. DNA Extraction from Plants Used in Transmission Trials

A DNA extraction protocol was used after being adapted from the Exgene™ Plant SV kit (Sunnyvale, CA, USA). Specifically, 0.1–0.15 g of fresh plant material was cut from periwinkle plants and weighed. The plant material was then placed into extraction bags, and 1.5 mL of PL buffer (cell lysis) was added. Only 1.5 mL of the macerated plant material was stored in a 1.5 mL Eppendorf tube. Next, 5 µL of RNAse (100 mg/mL) was added and vortexed. The tube was incubated at 65 °C for 30 min in a thermal block, with the inversion of the tube every 10 min. After that, 140 µL of PD buffer was added, the solution was vortexed, and the tube was incubated on ice for 5 min. The sample was then centrifuged for 5 min at 14,000 rpm at room temperature. The sample was transferred to a tube with columns and a filter and centrifuged again at 14,000 rpm for 2 min at room temperature. Next, 600 µL of the eluate was transferred to a 1.5 mL Eppendorf tube, and 900 µL of BD buffer (silica binding) was added. Then,700 µL of this mixture was transferred to a tube with a filter column. The tube was centrifuged for 30 s at 14,000 rpm at 24 °C. The eluate was discarded, and this step was repeated. Then, 700 µL of CW buffer (washing) was added to the column, and the tube was centrifuged for 30 s at 14,000 rpm at room temperature. The eluate was discarded again. The collection tube was dried with absorbent paper, and the column was reinserted. Then, 300 µL of CW buffer was added to the columns, and the tube was centrifuged for 3 min at 14,000 rpm at 24 °C. The eluate was discarded again, and the column was transferred to a 1.5 mL Eppendorf tube. Finally, 100 µL of AE buffer was added directly to the center of the column membrane, and the tube was incubated for 5 min at room temperature. The tube was then centrifuged at 14,000 rpm for 1 min at 24 °C. The precipitate was stored at −20 °C.

### 4.5. DNA Extraction from Psyllids

The captured insects were stored in 1.5 mL Eppendorf tubes with 70% ethanol. For DNA extraction, a cetyltrimethylammonium bromide (CTAB) buffer was used. Ethanol was removed, and the tubes were placed in a Speedback for 20 min at 48 °C to remove any ethanol residue. β-mercaptoethanol (0.3 M) was added to the CTAB buffer (10 µL of β-mercaptoethanol per 1 mL of buffer). The insects were macerated, individually, in 1.5 mL Eppendorf tubes using a pestle, adding 250 µL of CTAB buffer with β-mercaptoethanol. The samples were incubated for 1 h at 60 °C in a thermoregulator, shaking the tubes by inversion every 10 min. The samples were then centrifuged for 5 min at 10,000× *g* at 24 °C. The supernatant was transferred to a new 1.5 mL Eppendorf tube (180 µL), and we added 180 µL of chloroform: isopropanol 24:1 at −20 °C and vortexed the solution. After centrifugation for 20 min at 15,000× *g* at 4 °C, the aqueous phase (150 µL) was transferred to a new 1.5 mL Eppendorf tube. The transferred aqueous phase was treated with 150 µL of isopropanol at −20 °C and mixed by inversion before being stored at −20 °C for 12 h. The tubes were then centrifuged for 20 min at 15,000× *g* at 4 °C, and the supernatant was removed. The pellet was washed with 1000 µL of 70% ethanol, vortexed, and centrifuged for 5 min at 15,000× *g* at 4 °C. The supernatant was discarded, and the pellet was dried in the Speedback. The dried pellet was resuspended in 50 µL of TE (1X) buffer and stored at 4 °C for at least 2 h before being stored at −20 °C [56].

### 4.6. Molecular Detection of Phytoplasmas in Psyllids and Plants Used in Transmission Trials

Phytoplasma detection was performed through “nested PCR” with universal primers that amplify regions of the *16S rRNA* gene [57,58]. For direct PCR, the P1/P7 primer pair was used [59,60]. The direct PCR amplification product was diluted 1:10 for nested PCR with R16F2n/R2. Each PCR reaction was carried out using 1–1.5 µL of template DNA, 1 µL of each universal primer (25 µM), 3 µL of PCR buffer (10×), 1.5 µL of MgCl2 (50 µM), 1 µL of dNTPs (10 µM), 20.8 µL of ultra-pure water, and 0.2 µL of Platinum Taq polymerase enzyme (Invitrogen). All reagents were placed in a 200 µL thin-wall tube. Reactions were performed in a thermal cycler (Techne TC 3000 6) with an initial incubation cycle of 1 min at 94 °C, followed by 35 cycles, each consisting of denaturation for 30 s at 94 °C. We performed primer annealing for 45 s at 53 °C, elongation for 90 s at 72 °C, and final extension at 72 °C for 7 min.

The PCR amplification products were separated by agarose gel electrophoresis (1.2% agarose), stained with ethidium bromide, and run in TAE 1X buffer. A 100 bp “DNA Ladder” was used as a reference marker. Electrophoresis was carried out at 145 V for 25 min using a power supply (EPS—300 IIV). The amplification products were visualized under UV light using a UV transilluminator, and images of the gels were taken with a UV documentation system; then, the amplification products were sequenced as Psomagen (Rockville, MD, USA).

The obtained sequences were compared with those available in GenBank using BLASTn 2.16.0+ (25 June 2024). We performed bioinformatic analysis, including sequence alignments in Bioedit v7.2 and phylogenetic tree construction using neighbor-joining and maximum likelihood methods, using the MEGA v7.0 software.

Based on the sequences obtained from the *16S rRNA* gene amplicons, virtual restriction fragment length polymorphism (RFLP) analysis was performed using *RsaI*, *BstUI*, and *HhaI* restriction enzymes [22].

Additionally, as confirmation, PCR was performed using the specific primer for the ribosomal subgroup 16SrIII-J, which amplifies the gene encoding the membrane’s immunodominant protein A (*IdpA*, a group 16SrIII-specific membrane protein) with primers IdpAJ-F and IdpAJ-R [32]. Reactions were carried out in a thermal cycler (Techne TC 3000 6) with an initial incubation cycle of 3 min at 95 °C, followed by 35 cycles, each consisting of denaturation for 30 s at 95 °C, primer annealing for 30 s at 54 °C, elongation for 30 s at 72 °C, and final extension at 72 °C for 7 min. The PCR amplification products were separated by electrophoresis and sequenced.

### 4.7. Detection of the SAP54 Effector Associated with Virescence and Phyllody

In this study, we performed PCR to amplify the gene encoding of the SAP54 effector using specific primers designed using CLC Genomics Workbench v24.0.1 software: SAP54-IIIJ-F (5′–ATGTTTAAATCAAAAAAACAATT-3′) and SAP54-IIIJ-R (5′-GATTTAAAAGTGTTTTATAAG-3′). The SAP54 effector is known to induce phyllody formation in infected plants and plays an important role in the pathogenicity of phytoplasma 16SrIII-J [31]. The same PCR reaction conditions were used as were employed for the IdpAJ primers. The PCR amplification products were separated by electrophoresis.

### 4.8. COI Gene Amplification

The molecular identification of the insects was performed by amplifying the cytochrome c oxidase subunit I (*COI*) gene, known as the “DNA Barcode”, as these protein-coding genes are among the most conserved in the mitochondrial genome of animals [61]. The *COI* gene was amplified using the primers LCO1490 and HCO2198 [61]. The reactions were carried out in a thermocycler (Techne TC 3000 6) with an initial incubation cycle of 7 min at 94 °C, followed by 35 cycles, each consisting of 30 s of denaturation at 94 °C, primer annealing for 45 s at 46 °C, elongation for 70 s at 72 °C, and a final extension at 72 °C for 7 min.

The *COI* gene amplification products were sequenced by Psomagen (Rockville, MD, USA). The sequences obtained were compared to GenBank and the database of the Plant Virology and Bacteriology Laboratory at the University of Chile using BLASTn. This was followed by bioinformatics analysis including alignments in Bioedit v7.2 and phylogenetic tree construction using maximum likelihood method with the MEGA v7.0 program and the reference sequences of Appendix A.

### 4.9. Mitochondrial Genome Characterization

A single mallow psyllid was used to extract DNA. A total of 10 µL of DNA extract was loaded, and electrophoresis was performed on a 0.8% agarose gel stained with ethidium bromide using 1X TAE buffer. A 10 Kb “DNA Ladder” reference marker was used to determine the size of the amplified fragments. Electrophoresis was conducted with the power supply (EPS—300 IIV) at 140 volts for 30 min. Bands were visualized using a UV light transilluminator, and the obtained gels were photographed with a UV light photodocumentation system. The expected band size was 16,000 bp. DNA quantification was carried out using the EPOCH microplate spectrophotometer from BioTek (Winooski, VT, USA).

The genetic material was sequenced on the MGI Genomics platform, with an estimated total of 10 million reads of 150 bp in length. De novo assembly was performed using the CLC Genomics Workbench v24.0.1 platform. For gene annotation of the mitochondrial genes, the MITOS2 Web Server platform (http://mitos.bioinf.uni-leipzig.de/; accessed on 20 November 2024) was used with invertebrate mitochondrial genetic codes. The complete mitochondrial genome map of the mallow psyllid was constructed using CLC Genomics Workbench v24.0.1. The obtained sequence, which contains the mitochondrial genome, was compared with those in GenBank using BLASTn. Bioinformatic analysis was conducted, including alignments in BioEdit v7.2 and phylogenetic tree construction using maximum likelihood method in the MEGA v7.0 program and the reference sequences of Appendix A.

## 5. Conclusions

A cryptic species of the *Russelliana solanicola* complex, the mallow psyllid, is a vector of the phytoplasma 16SrIII-J. This has agronomic implications regarding the epidemiology and management of diseases caused by phytoplasmas. As a vector, the mallow psyllid expands the range of known vectors. It is probably part of a complex of cryptic species within *R. solanicola s.l.*, and it is currently unknown whether the other species within the complex, defined by their host plants, can transmit the same phytoplasmas. If this is the case, the potential for the disease to affect many crops of agronomic interest is high.

Here, a new strain of the phytoplasma 16SrIII-J has been described that lacks the SAP54 effector. Apparently, this new strain does not possess a fundamental effector, making it less pathogenic. This could be relevant from an evolutionary perspective, as the new strain could be losing effectors to thrive unnoticed in its hosts. Further studies are required to corroborate the reduction in pathogenicity of the phytoplasma 16SrIII-J strain infecting mallow plants.

## Figures and Tables

**Figure 1 plants-14-01279-f001:**
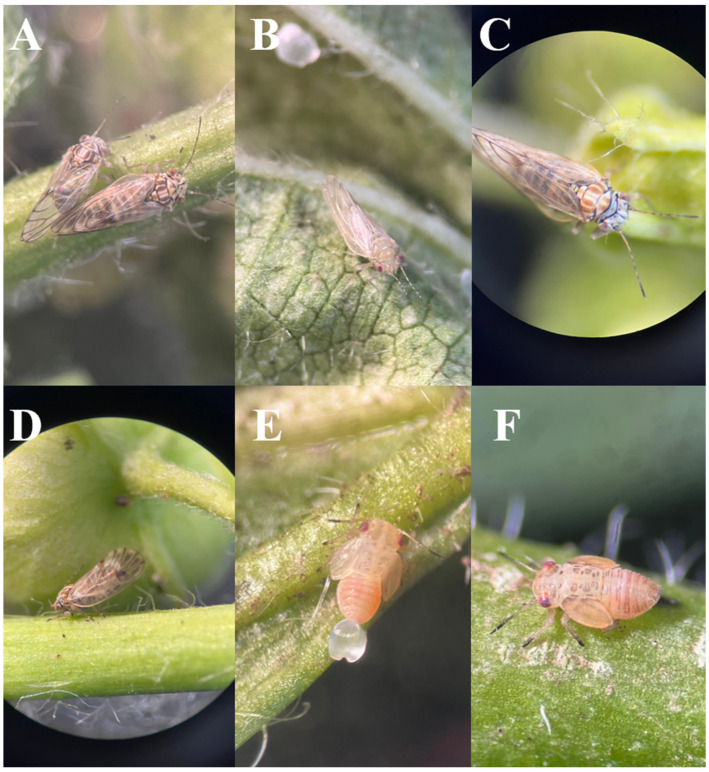
Mallow psyllid, morphologically identified as *Russelliana solanicola*. (**A**,**C**): full body, dorsal view; (**B**): newly emerged adult; (**D**): full body, lateral view; (**E**,**F**): immatures, full body, dorsal view.

**Figure 2 plants-14-01279-f002:**
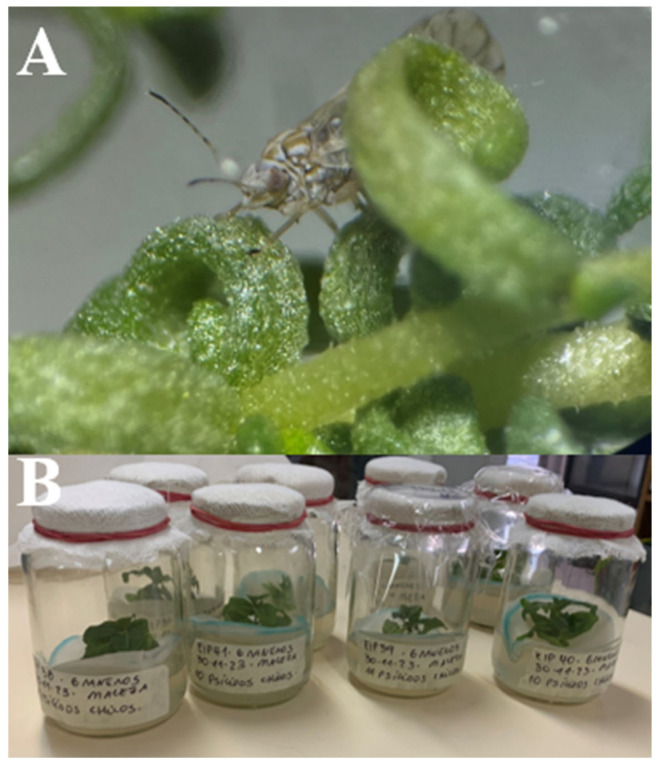
Transmission trials. (**A**): mallow psyllid in a jar during transmission trial; (**B**): transmission trial in jars with periwinkle plants (*C. roseus*).

**Figure 3 plants-14-01279-f003:**
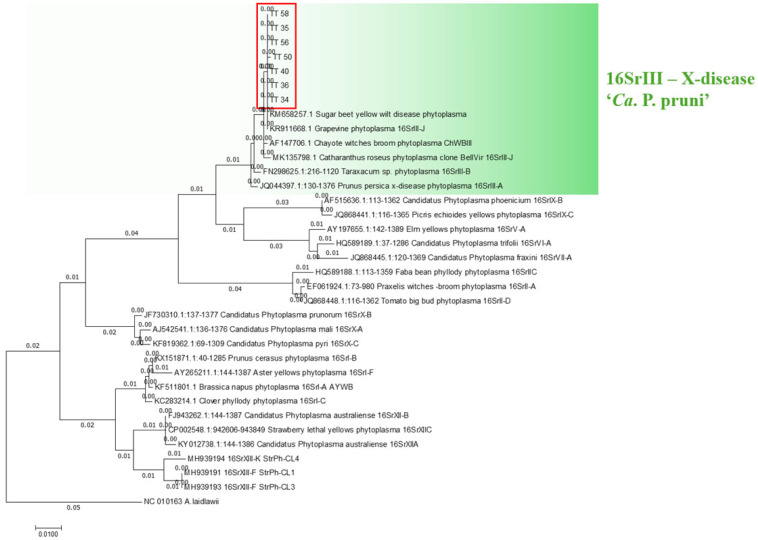
A phylogenetic tree constructed using the maximum likelihood method based on nucleotide sequences of the *16S rRNA* gene of the 16SrIII-J phytoplasma from periwinkle plants infected during transmission trials.

**Figure 4 plants-14-01279-f004:**
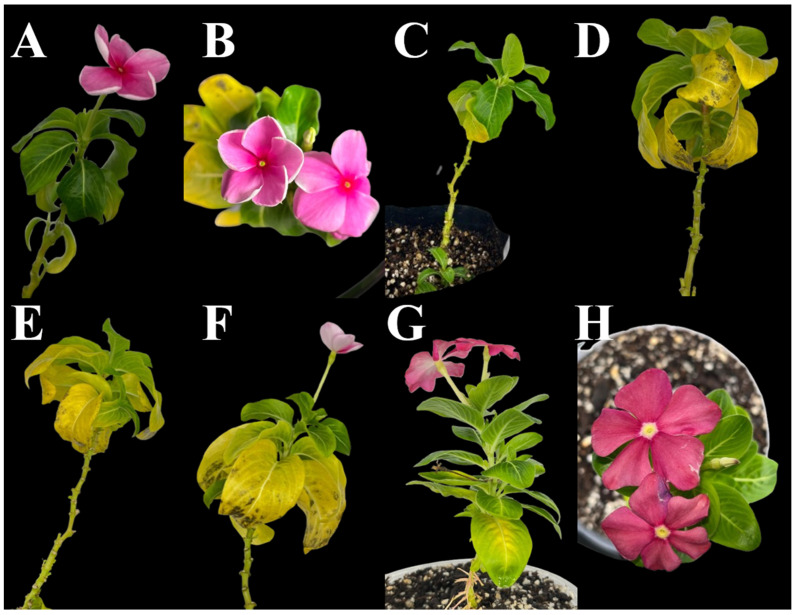
Periwinkle plant TT 34 infected by 16SrIII-J phytoplasma in different months after start of transmission test (**A**–**F**) never showed phyllody. (**A**): 6 months; (**B**): 7 months; (**C**): 8 months; (**D**,**E**): 9 months; (**F**): 10 months. (**G**,**H**): uninfected plants at the end of transmission trial (12 months).

**Figure 5 plants-14-01279-f005:**
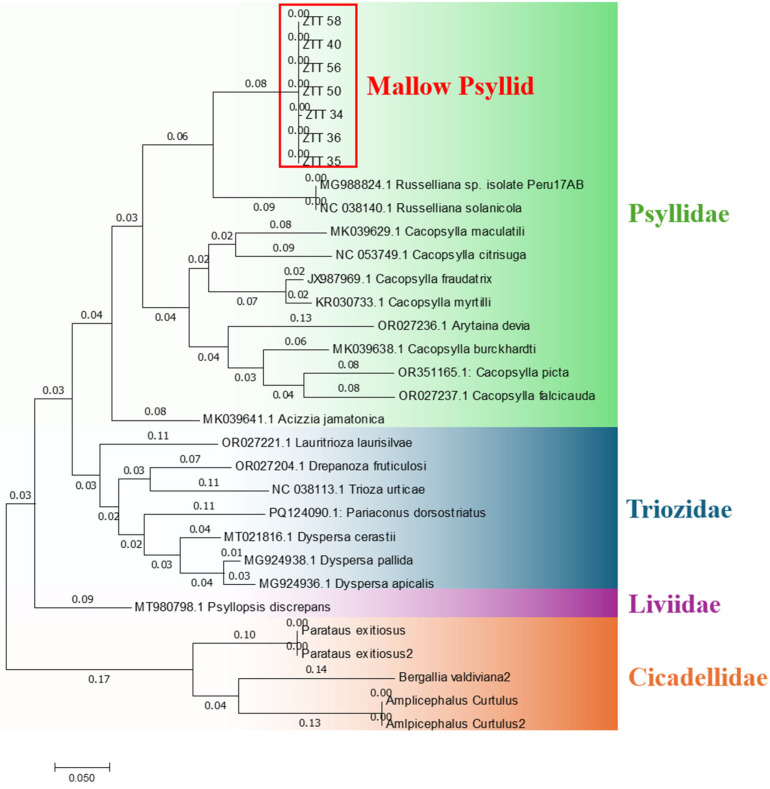
Phylogenetic tree using maximum likelihood method constructed based on nucleotide sequences of *COI* gene from captured and reference insects.

**Figure 6 plants-14-01279-f006:**
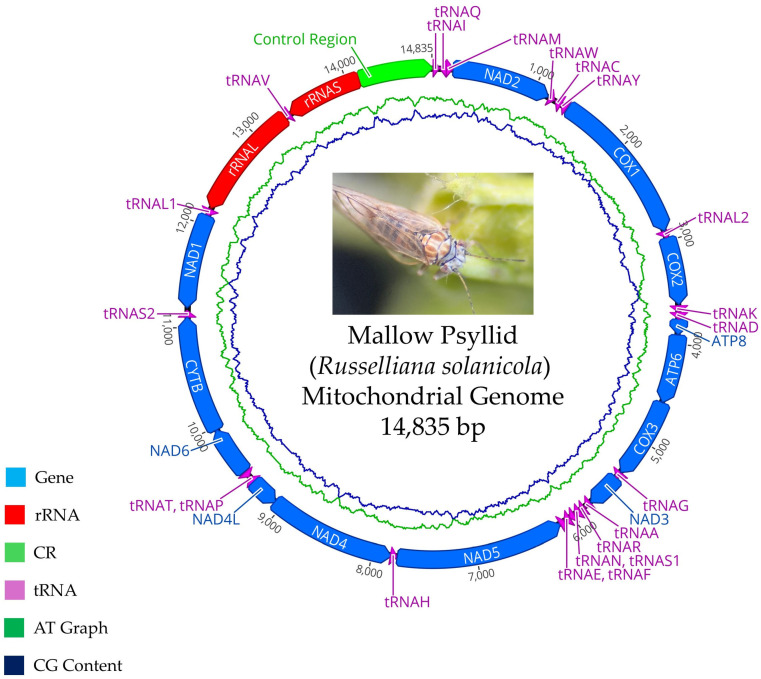
Map of complete mitochondrial genome of mallow psyllid.

**Figure 7 plants-14-01279-f007:**
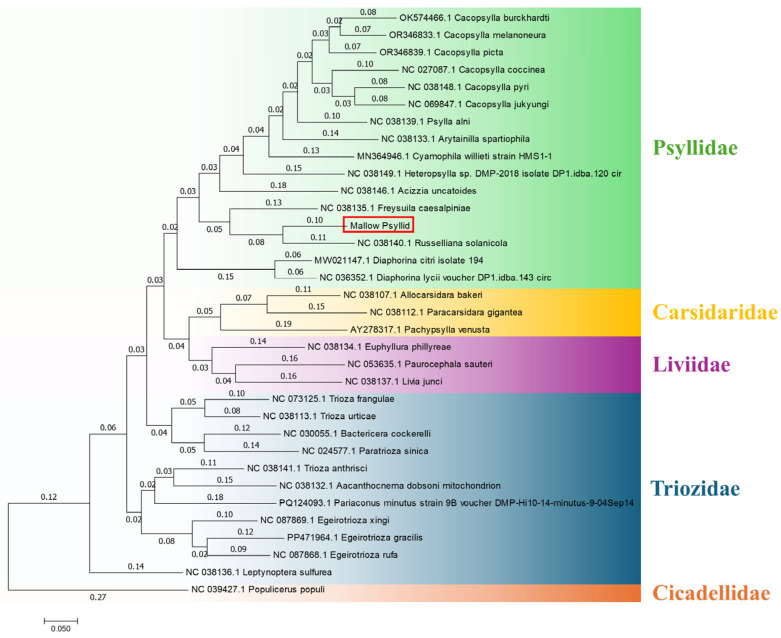
A phylogenetic tree constructed based on the maximum likelihood method using nucleotide sequences of the complete mitochondrial genome of mallow psyllid and reference insects.

**Table 1 plants-14-01279-t001:** Details of each field trip during which the specimens under study (mallow psyllid) were collected. The table specifies the number of insects sampled and the number of transmission trials (TTs) initiated.

Collection N°	Start Date of the Transmission Trial	Number of Immatures/Adults Collected	Number of Adults Used in Transmission Trials	TT Initiated
1	4 October 2023	15/100	98	14
2	17 October 2023	0/70	70	9
3	3 November 2023	0/50	44	4
4	16 November 2023	0/60	60	6
5	30 November 2023	12/105	90	9
6	27 December 2023	10/110	99	9
7	24 January 2024	0/80	80	8
8	7 February 2024	0/94	70	7
9	21 February 2024	14/115	96	8
10	6 March 2024	100/600	96	8
11	18 March 2024	58/200	320	16
12	5 April 2024	66/160	160	8
13	16 April 2024	0/50	50	5
14	17 May 2024	0/20	20	2
Total individuals		275/1814	1353	113

**Table 2 plants-14-01279-t002:** Transmission trial (TT) results on periwinkle plants. Phytoplasma detection using nested PCR for the *16S rRNA* gene. BLASTn from the amplification product obtained with the R16F2n/R2 primer pair. SBYWDP: *sugar beet yellow wilt disease phytoplasma*.

Phytoplasma-Positive Plant Code at the End of TT	TT Start Date	Months After the Start of TT in Which the Plant Was Detected Positive for 16SrIII-J	Closest Phytoplasma with BLASTn	Sequence % Identity	Accession Number
TT 34	30 November 2023	3	SBYWDP	100	PV053118
TT 35	30 November 2023	6	SBYWDP	99.82	PV053117
TT 36	30 November 2023	6	SBYWDP	100	PV053116
TT 40	30 November 2023	7	SBYWDP	99.82	PV053115
TT 50	27 December 2023	7	SBYWDP	100	PV053114
TT 56	24 January 2024	7	SBYWDP	100	PV053113
TT 58	24 January 2024	7	SBYWDP	99.91	PV053112

**Table 3 plants-14-01279-t003:** Nucleotide similarities obtained with *COI* mitochondrial gene sequences of insects that fed on phytoplasma-positive periwinkle plants.

Insect Code	Acc. Number	Closest species with BLASTn *COI*	% Coverage	% Identity	Acc. Number
ZTT 34	PV055068	*Russelliana solanicola*	99	84.84	PV055068
ZTT 35	PV055070	*Russelliana solanicola*	99	85.02	PV055070
ZTT 36	PV055069	*Russelliana solanicola*	99	85.02	PV055069
ZTT 40	PV055074	*Russelliana solanicola*	99	85.02	PV055074
ZTT 50	PV055071	*Russelliana solanicola*	99	85.02	PV055071
ZTT 56	PV055072	*Russelliana solanicola*	99	85.02	PV055072
ZTT 58	PV055073	*Russelliana solanicola*	99	85.02	PV055073

**Table 4 plants-14-01279-t004:** Nucleotide identities of mitochondrial genes of the mallow psyllid and *R. solanicola*.

Mallow Psyllid Gene	*R. solanicola* Gene	% Nucleotide Identity
Ins ZEIP gen 1 (*ND 1*)	*NAD 1*	80.74
Ins ZEIP gen 2 (*CYTB*)	*CYTB*	79.93
Ins ZEIP gen 3 (*NAD 6*)	*NAD 6*	75.61
Ins ZEIP gen 4 (*NAD 4*) Ins ZEIP gen 5 (*NAD 4L*)	*NAD 4* *NAD 4L*	78.80 84.38
Ins ZEIP gen 6 (*NAD 5*)	*NAD 5*	78.23
Ins ZEIP gen 7 (*NAD 3*)	*NAD 3*	78.16
Ins ZEIP gen 8 (*COX 3*)	*COX 3*	81.71
Ins ZEIP gen 9 (*ATP 6*) Ins ZEIP gen 10 (*ATP 8*)	*ATP 6* *ATP 8*	78.67 74.16
Ins ZEIP gen 11 (*COX 2*)	*COX 2*	83.13
Ins ZEIP gen 12 (*COX 1*)	*COX 1*	84.67
Ins ZEIP gen 13 (*NAD 2*)	*NAD 2*	76.83

**Table 5 plants-14-01279-t005:** Composition and annotation of mitochondrial genome of mallow psyllid. *nd: not determined.

Gene	Position	RNA Strand Orientation	Length (bp)	Initiation Codon	Termination Codon	Anticodon	Intergenic Nucleotides
*tRNAI*	1–63	+	63	*nd	nd	GAT	0
*tRNAQ*	61–127	−	67	nd	nd	TTG	−2
*tRNAM*	127–190	+	63	nd	nd	CAT	−1
*NAD2*	191–1159	+	969	ATA	TAA	nd	0
*tRNAW*	1163–1224	+	62	nd	nd	TCA	3
*tRNAC*	1227–1291	−	65	nd	nd	GCA	2
*tRNAY*	1293–1356	−	64	nd	nd	GTA	1
*COX1*	1372–2904	+	1533	ATG	TAA	nd	15
*tRNAL2*	2905–2969	+	65	nd	nd	TAA	0
*COX2*	2970–3627	+	657	ATG	TAA	nd	0
*tRNAK*	3634–3703	+	70	nd	nd	CTT	6
*tRNAD*	3704–3767	+	64	nd	nd	GTC	0
*ATP8*	3768–3920	+	153	ATT	TAA	nd	0
*ATP6*	3914–4588	+	675	ATG	TAA	nd	−7
*COX3*	4588–5370	+	783	ATG	TAA	nd	−1
*tRNAG*	5371–5432	+	62	nd	nd	TCC	0
*NAD3*	5433–5783	+	351	ATT	TAA	nd	0
*tRNAA*	5787–5848	+	62	nd	nd	TGC	3
*tRNAR*	5852–5913	+	62	nd	nd	TCG	3
*tRNAN*	5913–5976	+	64	nd	nd	GTT	−1
*tRNAS1*	5977–6029	+	73	nd	nd	GCT	0
*tRNAE*	6030–6090	+	61	nd	nd	TTC	0
*tRNAF*	6079–6141	−	63	nd	nd	GAA	−12
*NAD5*	6142–7762	−	1620	TTG	TAA	nd	0
*tRNAH*	7763–7825	−	63	nd	nd	GTG	0
*NAD4*	7821–9068	−	1248	ATG	TAA	nd	−5
*NAD4L*	9062–9349	−	288	TTG	TAG	nd	−7
*tRNAT*	9351–9412	+	62	nd	nd	TGT	1
*tRNAP*	9413–9474	−	62	nd	nd	TGG	0
*NAD6*	9478–9963	+	486	ATA	TAA	nd	3
*CYTB*	9963–11,099	+	1136	ATG	TAA	nd	−1
*tRNAS2*	11,099–11,161	+	63	nd	nd	TGA	−1
*NAD1*	11,187–12,101	−	915	ATA	TAA	nd	26
*tRNAL1*	12,102–12,166	−	65	nd	nd	TAG	0
*rRNAL*	12,172–13,318	−	1146	nd	nd	nd	5
*tRNAV*	13,315–13,375	−	61	nd	nd	TAC	−3
*rRNAS*	13,374–14,119	−	746	nd	nd	nd	−1
Control Region	14,120–14,835		735	nd	nd	nd	0

## Data Availability

All data are available in the manuscript and in Appendix A. Sequences can be accessed via Acc. Numbers.

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
