# Peer review of "First Report of a Psyllid Vector of ‘Candidatus Phytoplasma pruni’ (Strain 16SrIII-J)"

_plants, 2025, doi:10.3390/plants14091279_

Round 1

Reviewer 1 Report

Comments and Suggestions for Authors

The manuscript reports evidence that a possible cryptic species of the mallow psyllid is a vector of the phytoplasma 16SrIII-J, commonly transmitted by leafhoppers, which expands the range of known insect vectors, with epidemiological implications.

The study was well designed, the methodology is suitable for the study and the subject is relevant. Therefore, I suggest that the manuscript be accepted with minor edits. Please review the text for the correct use of the English vocabulary, typos and/or misspelling.

Is phytoplasma 16SrIII-J mainly transmitted by leafhoppers? Considering a transmission rate of 6.2% to the weed species, what is the economic importance of this phytoplasma? Why did you choose to run the transmission assays with a weed species (periwinkle plants)? Please clarify these points in the Introduction.

Tables and figures should be read without the need to go back to the main text. Please improve the title of tables and figures, so that they contain all the details about the assay (insect species, location, date, etc).

Lines 167-168: “The virtual RFLP pattern derived from the F2nR2 fragment of the 16S ribosomal DNA of the sample is identical”. What sample?

Line 178: “after digestion with, with,”

Lines 183-184: Again, do you refer to the plant or the insect here?

How many insect samples were analyzed by PCR? In the text, the authors mention that all insect samples were positive, but tables 3 and 5 show only one result.

Lines 215-217: please rewrite this sentence. It is confusing and the word “neither” was not appropriately used. Also, the sentence was not finished.

Please revise the text for the appropriate use of English grammar, vocabulary or typos. For example, in line 300, the correct would be insect vector. In line 306, I guess the authors intend to say “not the same species”. Lines 323-324: “there are no clear-cut morphological diferences”.

Lines 411-414 are repeated in the text. Indeed, this paragraph needs to be carefully reviewed.  

In the Conclusions section, the authors affirm that “the mallow psyllid is a vector of the phytoplasma 16SrIII-J”. Would it be safer to say that a cryptic species of the mallow psyllid is a vector of this phytoplasma? According to the Discussion section, it might be a different species as well.

Comments on the Quality of English Language

Please review the text for the correct use of the English vocabulary, typos and/or misspelling.

Reviewer 2 Report

Comments and Suggestions for Authors

This paper report on trasmission of 16SrIII-J phytoplasma strain through the mallow psyllid (Russelliana solanicola). The major content of this paper including molecular and morphological identification of the mallow psyllid R. solanicola is more appropriate for an entomological journal. In addition, further experimental work is required to prove the role of the psyllid R. solanicola as a vector of 16SrIII-J phytoplasma strain. No data were provided about acquisition access period (AAP), latency period (LP) and inoculation access period (IAP). Knowledge on these aspects is essential for elucidating the transmission process. Therefore, the paper is not suitable for publication in its current form.

Reviewer 3 Report

Comments and Suggestions for Authors

This paper provides an interesting report of a novel type of phytoplasma-vector interaction, i.e. 16SrIII-psyllids. The reported results are of certain importance for researchers studying phytoplasma transmission and epidemiology, opening to a complex scenario of multifaceted interactions. I must remark that, beside the sure interest of the work, the paper has several limitations in the presentation of methods and results, please find some suggestion to improve it.

Introduction - L54-55: This sentence is not completely clear: do you mean that the species other than Russelliana solanicola are monophagous/oligophagous? Please specify.

Results- figure 1-2: despite interesting, I believe that those figures may be moved to the supplementary materials, considering that there are so many figures. Instead, I suggest adding a table or a short list in the text to five some details about the trend of psyllid captures (abundance, collected stages, ….) over the collection season, as it may be relevant to assess transmission dynamics and disease epidemiology. This could be also incorporated into the current Table 1. For example, it wold be important to specify what insect instars were used for transmission trials.

Chapter 2.1 and 2.2. should be merged as they are both very short or even a single sentence.

Table 3. There is no need to prepare a table for a single line, please incorporate the information (sequence similarity + closest relative acc.nr.) in the text. The same is valid for Table 4, Table 5 and Table 9.

Figure 6 and 7 should be merged into a single panel as they basically show the main concept. Most importantly, in those figure an uninfected control is missing, so it is impossible to discriminate possible symptoms other than phyllody on infected plants, or to establish if they are totally asymptomatic. Please include the control. In figure 7 it is impossible to dechiper from the caption what the authors want to show, as pictures A, B, c are indicated as the same but they obviously focus on different aspects of the same TT.

Chapter 2.6: “All the samples that successfully amplified the mitochondrial COI gene” how many samples were analysed and how many did actually amplify? It is important to understand if some samples failed just for a casual PCR defect or if they correspond to a separate population that is not successfully amplified by the universal primers. Moreover, the low sequence similarity with the closest relative may be due to the fact that you have used Genbank, which is very limited and often imprecise to describe COI sequences. The BOLD database should be the correct reference in this case.

Discussion:

L364-365: “Since adult insects were used in all transmission trials, a long LP could even exceed the insect's lifespan” This sentence makes no sense to me, as if transmission was done by field collected insects so the time of possible acquisition in unknown. Of course LP may exceed the survival, but this is not correlated with the fact that you have collected adults.

I suggest adding a final paragraph in the discussion section as a final conclusion. For example, this paragraph could highlight the importance of identifying a new vector from the point of view of phytoplasma epidemiology. Even if the psyllid is not yet known to transmit a disease agent to crop plants, it could maintain the phytoplasma in plant reservoirs that could be finally exploited by polyphagous leafhoppers. In turn, this will also affect agronomic practice and preventive measures to be applied for disease management.

Materials and methods – chapter 4.2 Since the morphological traits to recognize the genus Russelliana are derived from the literature, there is no need to fully describe them. The morphological identification may be incorporated in the following paragraph just mentioning the related references.

L468: change “free phytoplasmas periwinkle plants” to “ phytoplasma-free periwinkles”. What was the survival rate of psyllids on periwinkle? could it be an issue for the low transmission rate?

Chapter 4.4: please provide the manufacturer detail for the kit that was used. Also, there is no need to specify all extraction steps, including those that were the same as described by the manufacturer’s instruction. Please only specify the steps that were customized here.

Chapter 4.7 Are the primers listed here a newly designed pair? In case please describe how they were designed (e.g. any software or tool that was used). If not please provide a reference.

Chapter 4.8 This is a single- sentence chapter. It should be moved just below the PCR describing 16S-based diagnosis.

chapter 4.9 I feel that too many unnecessary details are given here to describe a well established standard protocol. For example, there is no need to give the primer sequence as they are already described, furthermore the PCR thermal conditions should be provided only if they diverge from already described in the mentioned reference. There is no need to describe the full electrophoresis conditions. They are supposed to be the same as for all other PCR tests that have been described in the previous chapters. Finally, Table 10 could be moved to the supplementary materials.

Chapter 4.10; again, too many details are given about DNA electrophoresis prior to sequencing, on the other hand, no detail was given about the sequencing platform that was applied. Finally, also Table 11 should be moved to the supplementary materials, as only strictly necessary data should be indicated in figures and tables in the text.

Round 2

Reviewer 2 Report

Comments and Suggestions for Authors

Concerns were not properly addressed.

Author Response

Thank you very much for the suggestions.
Best regards

Reviewer 3 Report

Comments and Suggestions for Authors

The authors have provided suitable responses to my previous comments, and applied most of the changed that I had proposed.

Author Response

(The authors gave the same response as above.)
